# Facile Preparation of Samarium Carbonate-Polymethacrylate Microspheres as a Neutron-Activatable Radioembolic Agent for Hepatic Radioembolization

**DOI:** 10.3390/pharmaceutics15030877

**Published:** 2023-03-08

**Authors:** Yin How Wong, Azahari Kasbollah, Basri Johan Jeet Abdullah, Chai Hong Yeong

**Affiliations:** 1School of Medicine, Faculty of Health and Medical Sciences, Taylor’s University, Subang Jaya 47500, Malaysia; 2Medical Advancement for Better Quality of Life Impact Lab, Taylor’s University, Subang Jaya 47500, Malaysia; 3Medical Technology Division, Malaysian Nuclear Agency, Bangi 43000, Malaysia

**Keywords:** Samarium-153, radioactive microspheres, hepatic radioembolization, neutron activation, hepatocellular carcinoma

## Abstract

Radioembolization shows great potential as a treatment for intermediate- and advanced-stage liver cancer. However, the choices of radioembolic agents are currently limited, and hence the treatment is relatively costly compared to other approaches. In this study, a facile preparation method was developed to produce samarium carbonate-polymethacrylate [^152^Sm_2_(CO_3_)_3_-PMA] microspheres as neutron activatable radioembolic microspheres for hepatic radioembolization. The developed microspheres emits both therapeutic beta and diagnostic gamma radiations for post-procedural imaging. The ^152^Sm_2_(CO_3_)_3_-PMA microspheres were produced from commercially available PMA microspheres through the in situ formation of ^152^Sm_2_(CO_3_)_3_ within the pores of the PMA microspheres. Physicochemical characterization, gamma spectrometry and radionuclide retention assay were performed to evaluate the performance and stability of the developed microspheres. The mean diameter of the developed microspheres was determined as 29.30 ± 0.18 µm. The scanning electron microscopic images show that the spherical and smooth morphology of the microspheres remained after neutron activation. The ^153^Sm was successful incorporated into the microspheres with no elemental and radionuclide impurities produced after neutron activation, as indicated by the energy dispersive X-ray analysis and gamma spectrometry. Fourier transform infrared spectroscopy confirmed that there was no alteration to the chemical groups of the microspheres after neutron activation. After 18 h of neutron activation, the microspheres produced an activity of 4.40 ± 0.08 GBq.g^−1^. The retention of ^153^Sm on the microspheres was greatly improved to greater than 98% over 120 h when compared to conventionally radiolabeling method at ~85%. The ^153^Sm_2_(CO_3_)_3_-PMA microspheres achieved suitable physicochemical properties as theragnostic agent for hepatic radioembolization and demonstrated high radionuclide purity and ^153^Sm retention efficiency in human blood plasma.

## 1. Introduction

Liver cancer is the sixth most common cancer in the world and poses a major threat to human health [1]. It is among the top three causes of cancer death in 46 countries, and among the top five in 90 countries. Its incidence is predicted to increase by 55% between 2020 and 2040 [2]. Currently, surgical resection, liver transplantation, radiofrequency or microwave ablation, and transarterial embolization are the main therapeutic strategies for liver cancer [3,4]. Among these strategies, transarterial embolization including transarterial chemoembolization (TACE) and transarterial radioembolization (TARE) are recommended to treat patients with unresectable liver cancer or fail from other therapies. The treatments deliver microspheres containing chemotherapy or radioactive drugs to the targeted tumor(s) to block the blood supply as well as delivering chemotherapeutics or ionizing radiation to the tumors [5,6,7]. In addition, TARE is a preferred treatment for advanced-stage liver cancer with portal vein thrombosis and has been regarded as a promising treatment for intrahepatic cholangiocarcinoma [8,9].

Radioactive microsphere is an ideal radioembolic agent for TARE. The administered radioactive microspheres will be lodged at the terminal arterioles of the hepatic artery. Conventionally, the radioactive microspheres are prepared by conjugating or labeling radioactive isotopes onto a carrier such as glass, resin or polymeric particles [10,11,12,13]. Unfortunately, this mode of radioactive microspheres preparation requires the handling personnel to deal with highly radioactive materials during the preparation process. In addition, radioactive wastes may be produced during the isolation and purification of the radioactive microspheres. Furthermore, the radioactive microspheres often have to be prepared in higher activity to accommodate for the radioactive decay during production and transportation. Therefore, the conventional method of radioactive microspheres preparation requires extensive radiation shielding facilities and hence increased the production cost. Unfortunately, the current commercially available radioactive microspheres for hepatic radioembolization, Yttrium-90 (^90^Y)-labeled glass microspheres (TheraSphere^®^, Ottawa, ON, Canada) and resin microspheres (SIR-Spheres, St Leonards, Australia) are produced using this method [14].

Neutron activation is a promising method in reducing direct handling of radioactive material during the production of radioactive microspheres. In this mode of production, a stable isotope is first incorporated into the carrier and then activated to become radioactive by capturing a thermal neutron in a nuclear reactor. The activity of the radioactive microspheres can be manipulated by the amount of neutron activatable isotope in the formulation, neutron cross-section, thermal neutron flux, and irradiation time. With the wide availability of 224 operable research reactors across 53 countries currently [15], the production of neutron-activated radioactive microspheres for hepatic radioembolization can be done locally (or at neighboring countries) with reduced costs.

Several stable isotopes that can be neutron activated such as Samarium-152 (^152^Sm), Holmium-165 (^165^Ho) and Lutetium-176 (^176^Lu) are suitable to be used to prepare the neutron activatable radioactive microspheres. Generally, the beta energy-emitting radionuclides used in the radioembolization are having a linear energy ranging from 0.1 to 2.2 MeV. The beta radiation would induces the production of reactive oxygen species, thus leading to cellular apoptosis or breakage of the cellular DNA. The tissue penetration range of the beta emitter is greater than 1 mm, which effectively cover most solid tumors when they are embolized in the peripheral artery [16]. The ^153^Sm selected in this study is a promising theragnostic radionuclide due to its optimum half-life of 46.3 h, relatively high thermal neutron activation cross-section of 210 barns and emitting both therapeutic beta particles of 0.81 MeV (20%), 0.71 MeV (30%), 0.64 MeV (50%) and diagnostic gamma energies of 103 keV (28%). The beta energy emitted from ^153^Sm has a maximum tissue penetration range of up to 4.0 mm (average 0.8 mm), while the gamma rays are well suited for gamma scintigraphy imaging [17].

The ^152^Sm has also been widely used as a therapeutic agent in palliative treatment of bone metastasis and radiosynovectomy since several decades ago. It is also a popular radiotracer in pharmacoscintigraphy and gastrointestinal gamma imaging studies [18,19,20]. The therapeutic efficiency of the Sm-153 loaded microspheres has been assessed on liver tumor bearing rats. The tumor-bearing rats administered with the Sm-153-loaded microspheres showed good treatment response with no tumor observed at end of the 60 days study while tumor volumes in the control group were 24-fold larger compared to baseline [21]. Recently, a few attempts have been made to produce alpha particles-based agent for hepatic radioembolization. The alpha particles possess a higher linear energy transfer (LET) than beta particles, thus potentially offering more efficient and specific killing of tumors. However, the alpha particles are more suitable for smaller tumors as the alpha particles is having a shorter tissue penetrating range which limits its long-range “cross-fire” effect [16,22,23].

In this study, neutron-activatable samarium carbonate-polymethacrylate [^152^Sm_2_(CO_3_)_3_-PMA] were prepared by in situ formation of samarium carbonate [^152^Sm_2_(CO_3_)_3_] in pores of commercially available polymethacrylate (PMA) microspheres. The developed microspheres have a mean diameter of ~30 µm. The physicochemical characteristics, radionuclide purity and radionuclide retention efficiency of the developed ^153^Sm-microspheres were studied.

## 2. Materials and Method

### 2.1. Synthesis of ^152^Sm_2_(CO_3_)_3_-PMA Microspheres

Commercially available cross-linked PMA microspheres with sulphoisobutyl cation exchange groups (Fractogel EMD SO^3−^) were procured from Merck Millipore (MA, USA). The PMA microspheres were used after removing the preservative agents, ethanol and sodium chloride, by washing thoroughly with distilled water. The microspheres were filtered using filter paper and dried at 70 °C for 24 h.

To prepare the ^152^Sm_2_(CO_3_)_3_-PMA microspheres, 1 g of the dried PMA microspheres was added to 10 mL of 10% (*w*/*v*) samarium-152 chloride (^152^SmCl_3_) solution and incubated for 30 min to allow the binding of ^152^Sm^3+^ ions to the negatively charged sulphoisobutyl groups on the PMA microspheres. The unbound ^152^Sm^3+^ ions were removed by washing three times with distilled water. Then, the PMA microspheres were treated with 50 mL of 3.5% (*w*/*v*) sodium carbonate (Na_2_CO_3_) solution. The carbonate ions reacted instantly with the ^152^Sm^3+^ ions in the pores of the PMA microspheres, forming the insoluble ^152^Sm_2_(CO_3_)_3_. The ^152^Sm_2_(CO_3_)_3_-PMA microspheres were then washed three times with distilled water and filtered using filter paper. The collected microspheres were dried at 70 °C for 24 h.

### 2.2. Neutron Activation of ^152^Sm_2_(CO_3_)_3_-PMA Microspheres

The ^152^Sm_2_(CO_3_)_3_-PMA microspheres were sent for neutron activation in an open-pool TRIGA Mark II research reactor (General Atomics, CA, USA) located at the Malaysian Nuclear Agency, Bangi, Malaysia. Prior to neutron activation, the microspheres were prepared following the same procedures, as described earlier [24]. Two neutron activation protocols, namely, pneumatic transfer system (PTS) and rotary specimen rack (RR), were used to neutron activate the samples. For the PTS protocol, the thermal neutron flux was set at 5 × 10^12^ n·cm^−2^·s^−1^ for a duration of 5 min. Meanwhile, the thermal neutron flux and duration for RR protocol were set at 2 × 10^12^ n·cm^−2^·s^−1^ and 6 or 18 h, respectively [24,25].

### 2.3. Radioactivity Assay

A calibrated radiopharmaceutical activity ionization chamber (CRC-25R, Capintec, NJ, USA) was used to measure the radioactivity of the neutron activated samples. The specific activity of the samples (GBq·g^−1^) was then determined from the radioactivity measured. The activity per microsphere (Bq per microsphere) were calculated using the following equation:(1)$$\mathrm{Activity}\;\mathrm{per}\;\mathrm{microsphere}=\frac{Specific\;activity\;of\;microspheres\;(GBq\cdot g^{-1})}{Number\;of\;microspheresper\;gram}$$

### 2.4. Gamma Spectroscopy

Gamma spectroscopy was carried out on the samples at 24 h and 48 h after neutron activation to investigate if there were any long-lived radionuclide impurities present in the samples. The gamma spectroscopy was performed using a hyper-pure germanium detector (Canberra, Meriden, CT, USA) and gamma spectrum analysis software (Genie^TM^ 2000 Ver. 3.2, Canberra, Meriden, CT, USA). Each sample was counted for 5 min at a distance of 18 cm from the detector so that the detection yield was less than 5% for minimal dead time effect.

### 2.5. Field Emission Scanning Electron Microscopy and Energy-Dispersive X-ray Spectroscopy

Field emission scanning electron microscopy (FESEM) was performed on a FESEM system (Quanta FEG 450, FEI, OR, USA) to monitor the shape and morphology of the ^152^Sm_2_(CO_3_)_3_-PMA microspheres before and after neutron activation. The SEM images were obtained at 5 kV, 10 mm working distance and spot size of 2.0 after mounting the samples on aluminum stubs. Elemental compositions of the ^152^Sm_2_(CO_3_)_3_-PMA microspheres were investigated using the energy dispersive X-ray (EDX) spectroscopy on the same FESEM system. The ^152^Sm content in the microspheres was determined using EDX spectroscopy. 

### 2.6. Particle Size Analysis

The laser scattering particle size distribution analyzer (Microtrac X100, Honeywell, USA) was used to determine the mean size and size distribution of the microspheres. Suspension of ^152^Sm_2_(CO_3_)_3_-PMA microspheres in distilled water was loaded into the particle size analyzer for the size measurements.

### 2.7. Fourier Transform Infrared Spectroscopy

The effect of neutron activation on the different chemical groups of the ^152^Sm_2_(CO_3_)_3_-PMA microspheres was assessed using Fourier transform infrared (FTIR) spectroscopy within 500–4000 cm^−1^ (Spectrum 100, PerkinElmer Inc., Waltham, MA, USA). The spectra of the samples before and after neutron activation were compared.

### 2.8. Thermogravimetric Analysis

The thermogravimetric analysis (TGA) profiles of the PMA microspheres and ^152^Sm_2_(CO_3_)_3_-PMA microspheres were analyzed using a thermogravimetric analyzer (TGA 8000™, PerkinElmer Inc., Waltham, MA, USA). About 5 mg of sample was added to the ceramic sample pan and scanned from 30 °C to 1000 °C at a scanning rate of 10 °C per min under constant nitrogen flow. Analysis of the TGA spectrum obtained were performed using the Pyris™ software (PerkinElmer Inc., Waltham, MA, USA).

### 2.9. Density and Viscosity Measurement

The density, *ρ*_s_ of ^152^Sm_2_(CO_3_)_3_-PMA microspheres was measured using helium gas pycnometer (AccuPvc II 1340, Micromeritics Ins. Corp., GA, USA) at 25 °C. The number of microspheres in one gram of the samples was estimated using the following equation:(2)$$\mathrm{Number}\;\mathrm{of}\;\mathrm{microspheres}\;\mathrm{per}\;\mathrm{gram}=\frac{6\times 10^{12}}{\pi \times {{\rho_{s}\times D}_{p}}^{3}}$$
where

*D_p_* = mean diameter of the microspheres (µm)

*ρ_s_* = microspheres density (g.cm^−3^)

The viscosity, *η*_0_, of 2.5% (*w*/*v*) microspheres in saline solution was measured at 37 °C, using HAAKE™ MARS™ Rheometer (Thermo Fisher Scientific Inc., Waltham, MA, USA). The value was then incorporated into Stokes’ Law (Equation (4)) to determine the sedimentation rate (settling velocity) of the suspension.
(3)$$V_{sed}=\frac{g{D_{p}}^{2}(\rho_{s}-\rho_{f})}{18\eta_{0}}$$
where

*V_sed_* = sedimentation rate (cm.s^−1^)

*g* = gravitational acceleration constant (981 cm·s^−2^)

*η*_0_ = dynamic viscosity of the fluid (P = g·cm^−1^·s^−1^)

### 2.10. In Vitro Radionuclide Retention Efficiency

The microspheres (0.75 g per vial) were neutron activated using PTS method for 5 min and divided equally into three 15 mL test tubes containing 10 mL of distilled water. Using a roller mixer (Movil-Rod, J.P. Selecta, Barcelona, Spain), the test tubes were mixed at 50 rpm for 1 h. Then, the microspheres were precipitated at 2000 rpm for 5 min and 1 mL of supernatant was transferred into gamma assay tubes. The extraction of the supernatant was repeated until a total of 8 mL supernatants was obtained from each sample over a period of 120 h. All supernatant samples were assayed using a gamma scintillation counter (2470 Wizard2, PerkinElmer Inc., Waltham, MA, USA). The experiment was repeated by replacing the distilled water with saline solution (0.9% NaCl) and human blood plasma. The radionuclide retention efficiency of the samples was calculated using Equation (4) [18].
(4)$$\mathrm{Retention}\;\mathrm{Efficiency}\;\left(\%\right)=\frac{\left(A_{\mathrm{sus}}-A_{\sup}\right)}{A_{\mathrm{sus}}}\;\times \;100\%$$
where A_sus_ is the activity of Sm_2_(CO_3_)_3_-PMA microsphere suspension before each extraction of 1 mL supernatant and A_sup_ is the activity of 1 mL supernatant.

## 3. Results

Neutron-activatable ^152^Sm_2_(CO_3_)_3_-PMA microspheres with high radionuclide retention efficiency were successfully produced in this study using a facile, efficient and inexpensive method. The microspheres synthesized were neutron activated using both PTS and RR protocols in a research reactor. The mean specific activity produced by PTS (5 min), RR (6 h) and RR (18 h) were 0.086 ± 0.009, 1.27 ± 0.13 and 4.40 ± 0.08 GBq·g^−1^, respectively. Four photopeaks (103, 69, 47 and 41 keV) were noticed in the gamma spectrum of the ^153^Sm_2_(CO_3_)_3_-PMA microspheres (Figure 1). The principal gamma energies emitted by ^153^Sm were responsible for the two prominent peaks at 103 and 69 keV while the photopeaks at 47 and 41 keV were due to K-shell characteristic X-rays following internal conversion. No long-lived radionuclide impurity was observed in the ^153^Sm_2_(CO_3_)_3_-PMA microspheres after neutron activation.

Figure 2 shows the SEM images of the Sm_2_(CO_3_)_3_-PMA microspheres before and after neutron activation. The Sm_2_(CO_3_)_3_-PMA microspheres were spherical and had a smooth surface, both before and after neutron activation. The structural integrity of the ^153^Sm_2_(CO_3_)_3_-PMA microspheres remained stable after 6 and 18 h neutron activation (Figure 2). Figure 3 shows the size distribution of the Sm_2_(CO_3_)_3_-PMA microspheres before and after neutron activation. The mean diameter of the ^152^Sm_2_(CO_3_)_3_-PMA microspheres was 29.3 ± 0.18 µm. No size reduction was observed after 6 and 18 h neutron activation. Overall, the size distribution of Sm_2_(CO_3_)_3_-PMA microspheres was remained within 20–40 µm both before and after neutron activation.

The EDX spectrum of the Sm_2_(CO_3_)_3_-PMA microspheres before and after neutron activation (Figure 4) indicated the presence of chemical elements carbon (C), oxygen (O), samarium (Sm) and sulphur (S), corresponding to the chemical elements present in the PMA microspheres and ^152^Sm_2_(CO_3_)_3_. Due to the limitation of EDX spectroscopy to detect elements with atomic numbers lower than 6, the element hydrogen (H) was not detected in the spectrum. No residual sodium (Na) or chloride (Cl) from the reagents [SmCl_3_ and Na_2_(CO_3_)] used during preparation were found in the samples. In accordance with the gamma spectrometry, no elemental impurity was observed in the ^153^Sm_2_(CO_3_)_3_-PMA microspheres. The EDX spectroscopy indicated ~3.5% of ^152^Sm in the ^152^Sm_2_(CO_3_)_3_-PMA microspheres.

Figure 5 shows the FTIR spectra of the ^152^Sm_2_(CO_3_)_3_-PMA microspheres and PMA microspheres. The presence of the sulphonate groups (-SO_3_^−^) on the PMA and ^152^Sm_2_(CO_3_)_3_-PMA microspheres were indicated by the presence of peaks within 1000–2000 cm^−1^. These peaks remained in the ^152^Sm_2_(CO_3_)_3_-PMA microspheres after neutron activation (Figure 5C), suggesting that the sulphonate groups, which are important to attract the positive charged ^152^Sm ions, remained intact during the neutron activation. In addition, no alteration to other functional groups of the microspheres was observed after neutron activation.

The TGA profiles of the PMA and ^152^Sm_2_(CO_3_)_3_-PMA microspheres are given in Figure 6. Both PMA and ^152^Sm_2_(CO_3_)_3_-PMA microspheres showed two major events of weight loss starting at 250 °C and above 700 °C owing to the pyrolysis of all carbon- and sulphur-containing compounds, respectively. The residual weight shown on TGA profile of ^152^Sm_2_(CO_3_)_3_-PMA microspheres indicated the presence of ^152^Sm metal in the sample. In accordance with the result obtained using EDX spectroscopy, the percentage of the ^152^Sm content in the ^152^Sm_2_(CO_3_)_3_-PMA microspheres was found at ~3.5%.

The physicochemical characteristics of the ^152^Sm_2_(CO_3_)_3_-PMA microspheres are listed in Table 1. The density of the ^152^Sm_2_(CO_3_)_3_-PMA microspheres, as determined by gas pycnometer was 1.3463 ± 0.0005 g·cm^−3^ (Table 1). There was an average of 56 million microspheres in one gram of ^152^Sm_2_(CO_3_)_3_-PMA microspheres. The activity per microsphere was 22.6 Bq and 78.6 Bq after 6 and 18 h neutron activation, respectively. The dynamic viscosities of the ^152^Sm_2_(CO_3_)_3_-PMA microsphere suspension at different shear rates are shown in Figure 7. There was only a small variation in the viscosity values obtained at different shear rates and thus a mean viscosity of 0.01067 ± 0.0003 g·cm^−1^·s^−1^ was used to determine the settling velocity for the ^152^Sm_2_(CO_3_)_3_-PMA microsphere suspension (Table 1). The estimated settling velocities were found to be 0.0149 ± 0.00005 cm·s^−1^.

The in situ formation of ^152^Sm_2_(CO_3_)_3_ in the pores of the PMA microspheres greatly reduced the leaking of radioactive ^153^Sm from the ^153^Sm_2_(CO_3_)_3_-PMA microspheres in distilled water, saline solution and blood plasma (Figure 8). The retention efficiencies of the ^153^Sm on the ^153^Sm_2_(CO_3_)_3_-PMA microspheres were ~99% in distilled water and saline solution, and ~98% in blood plasma over a period of 120 h.

## 4. Discussion

This study aimed to develop a facile and efficient method for preparing the neutron-activatable radioactive microspheres for hepatic radioembolization through in situ formation of Sm_2_(CO_3_)_3_ in the pores of the commercially available PMA microspheres. The preparation workflow described in this study can be performed in any standard pharmaceutical or chemical laboratory as no ionizing radiation is involved. The negatively charged groups on the PMA microspheres first bind the positively charged ^152^Sm^3+^ ions and then react with the CO_3_^2−^ ions to form the insoluble ^152^Sm_2_(CO_3_)_3_ complex in the pores of the microspheres.

An open-pool-type TRIGA MARK II reactor that operated at 750 kW power and 2.0–5.0 × 10^12^ n·cm^−2^·s^−1^ neutron flux was used in this study to neutron activate the stable ^152^Sm_2_(CO_3_)_3_-PMA microspheres. Long-duration RR irradiation method is more suitable for this type of reactor if high specific activity is desired. Depending on the neutron flux and amount of ^152^Sm in the sample, the irradiation time can be determined using the standard neutron activation equation. In this study, therapeutic activity close to 4.5 GBq.g^−1^ was produced from 18 h neutron activation. This is a relatively cheaper solution as natural abundant ^152^Sm, which contains only ~27% ^152^Sm was used. In order to achieve higher specific activity or to reduce irradiation time by 3–4 times, enriched-form ^152^Sm (>98% abundance) can be used; however, the only drawback is the cost which could be 250 times higher than the natural ^152^Sm.

The neutron activation of the microspheres has not altered the physicochemical properties of the ^152^Sm_2_(CO_3_)_3_-PMA microspheres. The microspheres remained spherical and smooth with no change in the particle size distribution even after 18 h neutron activation. This indicates that the developed microspheres tolerated high temperatures and harsh environments well during the long-duration neutron activation process. This is further supported by the TGA results showing that physical structure of the microspheres remained stable up to 250 °C. The FTIR results also showed chemical stability of the microspheres as the neutron activation has not produced any changes to the chemical groups of the microspheres. In summary, the microspheres did not receive any physical or chemical damage despite long irradiation time, which was not possible to achieve with biodegradable polymers such as poly(L-lactic acid) [26]. In addition, the microspheres produced in this study represent an improvement on the irregular shaped ^153^Sm-labeled Amberlite microspheres produced in earlier study [24]. Neutron activation of the ^1153^Sm-labeled Amberlite microspheres experienced fragmentation with reduction in the particle size after neutron activation.

During hepatic radioembolization, the radioembolic microspheres are suspended at a concentration of 2.5% (*w*/*v*) for intraarterial delivery. Recently, the European Association of Nuclear Medicine (EANM) has recommended suspending microspheres in 5% glucose solution to reduce stasis and reduces stasis and thereby procedural patient discomfort [27]. The density of the microspheres is an important parameter, as it determines the settling velocity of the microspheres in the microcatheter as well as blood vessels. Lower settling velocity of the microspheres is required to ensure even distribution of the microspheres within the tumor volume and to prevent premature settling of the microspheres in the microcatheter or blood vessels before reaching the tumor. In this study, the density of the ^153^Sm_2_(CO_3_)_3_-PMA microspheres was found to be comparable to the resin-based SIR-Spheres (1.6 g·cm^−3^) and much lower than the glass-based Therasphere (3.2 g·cm^−3^) [24]. The settling velocity of the ^153^Sm_2_(CO_3_)_3_-PMA microspheres was also relatively lower than the resin-based microspheres and much lower than the glass-based microspheres [28,29,30].

The in situ formation of the insoluble ^152^Sm_2_(CO_3_)_3_ in the pores of the PMA microspheres has greatly reduced the leaching of the radioisotope. Although the PMA microspheres used in this study have a negatively charged group that could be used to bind the positively charged ^152^Sm^3+^ ions through electrostatic interactions, the interaction between the ^152^Sm^3+^ ions and PMA microspheres was insufficient to prevent leaching of the radioactive ^153^Sm ions in both saline solution and human blood plasma. The presence of other positively charged ions in the saline solution and blood plasma could displace the ^152^Sm^3+^ ions and hence leading to the leaking of ^152^Sm^3+^ ions from the microspheres. There was about 3–15% leaching of ^153^Sm from the PMA microspheres observed in our preliminary experiments and was evident from a previous study [25]. The in situ formation of Sm_2_(CO_3_)_3_ in the pores of the PMA microspheres has increased the retention efficiency to be more than 98% in both saline solution and human blood plasma. The insoluble Sm_2_(CO_3_)_3_ has greatly prevented the displacement of Sm^3+^ ions from the negatively charged group by other positive charged ions present in the saline solution or blood plasma. High retention efficiency is important to prevent delivery of free ^153^Sm in healthy tissues.

Currently, the commercially available radioembolic agent for hepatic radioembolization are based on ^90^Y. ^90^Y is a pure beta emitter and hence post-administration imaging becomes a challenge. Alternative methods, such as Bremsstrahlung imaging and Positron Emission Tomography (PET) have been used to verify the treatment outcome; however, these techniques suffer from poor spatial resolution and sensitivity, respectively [31]. A recent study has shown that ^90^Y-based PET imaging could provide good imaging quality with the use suitable algorithm such as local deposition algorithm. However, the accurate absolute PET calibration will need to be performed and image acquisition times will need to be sufficiently long [32]. In view of the above, a theragnostics radionuclide that emits both high-energetic beta and low-energetic gamma radiations is ideal for solving this shortcoming of ^90^Y. Several radioisotopes, such as Rhenium-186 (^186^Re), Rhenium-188 (^188^Re), Holmium-166 (^166^Ho) and Samarium-153 (^153^Sm), have been evaluated and tested for this purpose [25,26,33,34]. Neutron activation of ^185^Re is not recommended as it produces both ^186^Re and ^188^Re via (n, 2n) and (n, ϒ) reactions. The isolation of ^188^Re is difficult due to its relatively shorter half-life than ^186^Re (17 h versus 3.7 days). Furthermore, ^186^Re decays to an unstable daughter nuclide with long physical half-life [17,35,36]. The relatively lower gamma energy of ^166^Ho maybe not sufficient to achieve high-resolution gamma imaging. In addition, both the ^166^Ho and ^188^Re possess low cross-section values of less than 100 barns [17,35,36]. In our opinion, ^153^Sm has an ideal physical half-life and emits both diagnostic gamma and therapeutic beta radiation renders it a suitable alternative radionuclide for hepatic radioembolization. In addition, ^153^Sm can be neutron activated from the stable parent (^152^Sm, natural abundance ~27%) and the decay product is the stable ^153^Eu, thus minimizing the radioactive waste production.

Neutron activation is a relatively straightforward and cost-effective method for radiotherapeutics production. Despite several advantages of neutron activation, production of radiotherapeutics using the neutron activation method has not yet been widely applied. This is mainly due to the significant amount of heat produced during the neutron activation process, which may degrade the carrier of the radiotherapeutics [26,37]. In order to overcome this limitation, the carrier made of PMA material was used in this study due to its high thermal stability [38]. In addition, the PMA polymer has excellent biological stability and resistance against neutron activation [38,39]. It is also biocompatible, resistant to physical heat, inert to physiological biochemicals, near blood plasma density and easily conjugate with the targeted radioisotope. The PMA microspheres used in this study is readily available in particle size of about 30 µm, which is the optimum size for hepatic radioembolization.

A simple and inexpensive method via in situ formation of ^152^Sm_2_(CO_3_)_3_-PMA microspheres was developed in this study. The synthesis method described in this study avoids the phosphate ions which is commonly used to form insoluble complexes. The ^90^Y-labeled microspheres (SIR-Spheres, Australia) precipitate the ^90^Y as phosphate salts in the microspheres during the preparation [40]. However, the phosphate salts are not compatible with neutron activation production on radiopharmaceuticals, as the phosphorus can be neutron activated to from radioactive phosphorus-32 with a physical half-life of 14.29 days [41,42].

The method described in this study can be used in the preparation of a wide range of neutron-activated radiopharmaceuticals, such as Ho-166 and Lu-177, but further investigation is needed to study their physicochemical characteristics. ^153^Sm-PMA microspheres were successfully developed as a potential theragnostic agent for hepatic radioembolization in this study. Further studies are needed to access the cytotoxicity and anticancer properties of the developed microspheres in comparison to other commercially available microsphere formulation in animal models.

## 5. Conclusions

^153^Sm-PMA microspheres were successfully developed as a potential theragnostics agent for hepatic radioembolization. The formulation was prepared using simple and inexpensive in situ formation of ^152^Sm_2_(CO_3_)_3_ in the pores of the PMA microspheres. The formulation can be activated in a thermal neutron flux of a research reactor operates at medium to high power (>10^12^ n·cm^−2^·s^−1^). The ^153^Sm_2_(CO_3_)_3_-PMA microspheres have a mean diameter of 29.3 ± 0.18 µm. The physicochemical properties of the microspheres were not altered by long duration (tested until 18 h) neutron activation. No radionuclide impurity was detected, and the radioactivity retention achieved ~99% and ~98% in saline solution and blood plasma, respectively. In conclusion, the ^153^Sm_2_(CO_3_)_3_-PMA microspheres prepared using a simple and inexpensive in situ formation method has met all the desired physicochemical characteristics of a theragnostics agent for hepatic radioembolization.

## Figures and Tables

**Figure 1 pharmaceutics-15-00877-f001:**
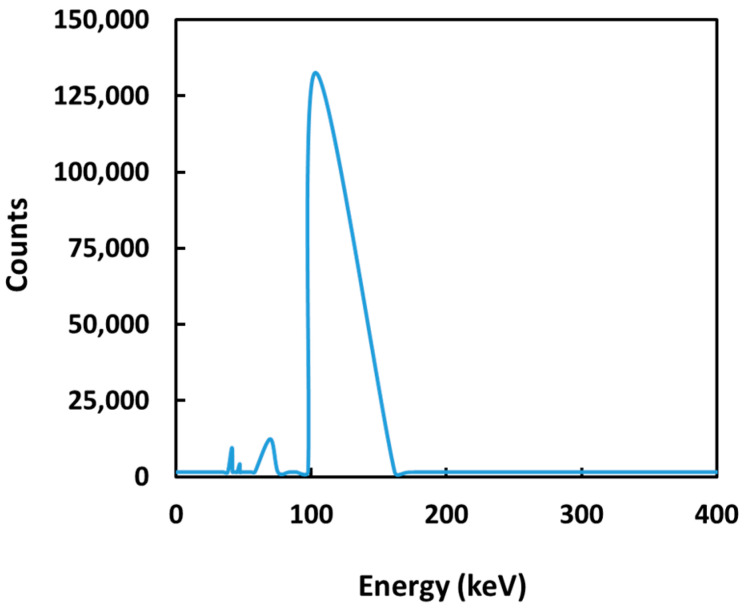
Gamma spectrum of ^153^Sm_2_(CO_3_)_3_-PMA microspheres at 24 h after neutron activation.

**Figure 2 pharmaceutics-15-00877-f002:**
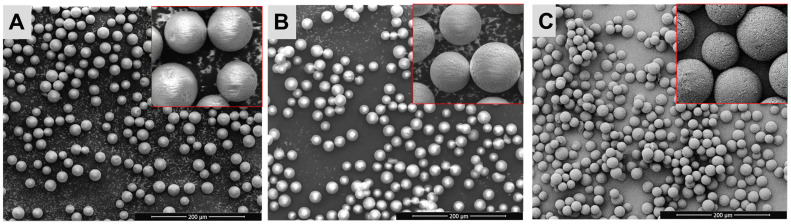
Scanning electron images of the ^152^Sm_2_(CO_3_)_3_-PMA microspheres both before (**A**) and after 6 h (**B**) and 18 h (**C**) neutron activation.

**Figure 3 pharmaceutics-15-00877-f003:**
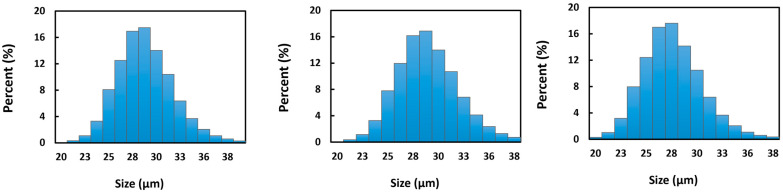
Particle size distribution of ^152^Sm_2_(CO_3_)_3_-PMA microspheres before (**A**) and after 6 h (**B**) and 18 h (**C**) neutron activation.

**Figure 4 pharmaceutics-15-00877-f004:**
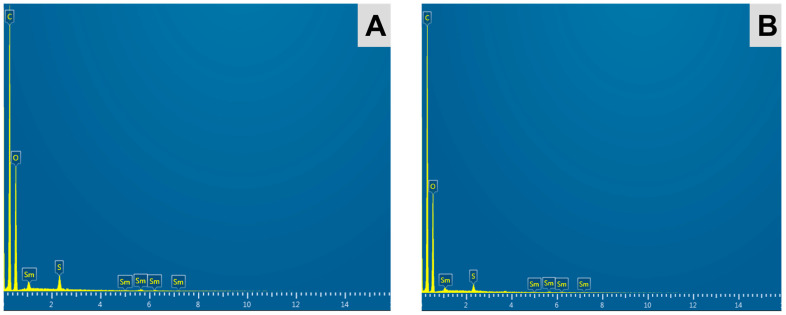
EDX spectra of the ^152^Sm_2_(CO_3_)_3_-PMA microspheres both before (**A**) and after neutron activation (**B**), showing the presence of carbon (C), oxygen (O), sulphur (S) and samarium (Sm) elements in the samples.

**Figure 5 pharmaceutics-15-00877-f005:**
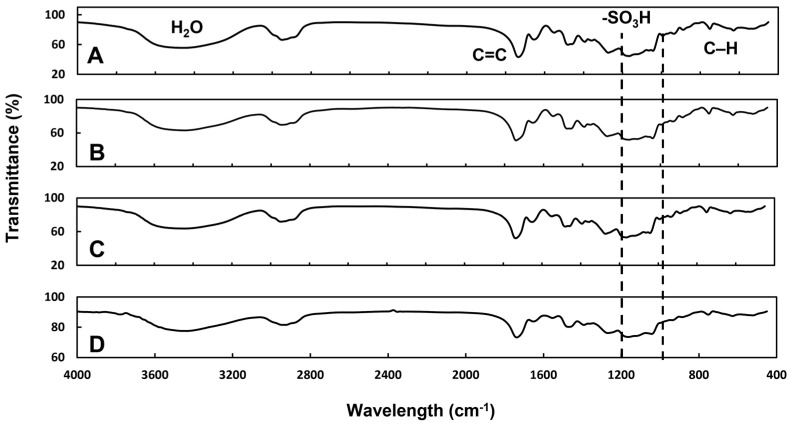
FTIR spectrum of the PMA microspheres (**A**), ^152^Sm_2_(CO_3_)_3_-PMA microspheres both before and after neutron activation (**B**,**C**), samarium carbonate (**D**).

**Figure 6 pharmaceutics-15-00877-f006:**
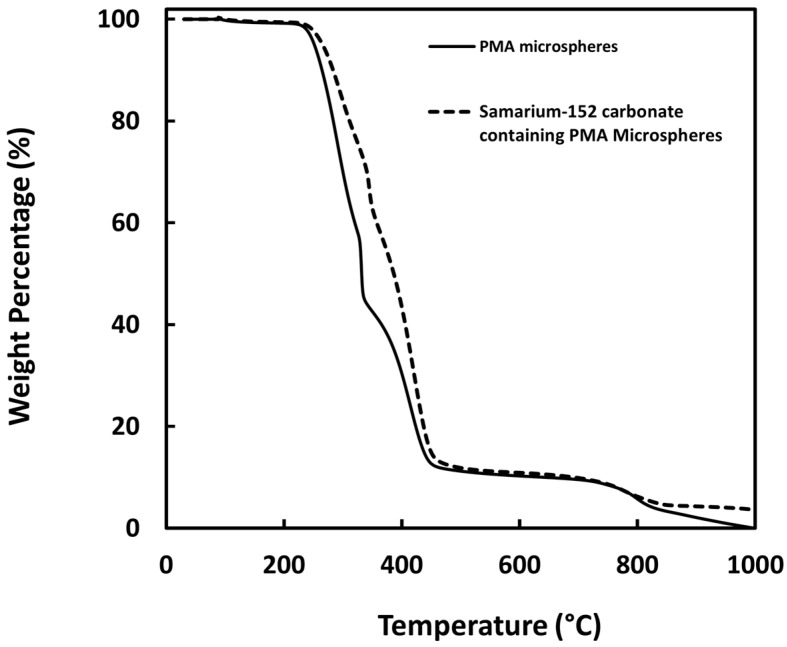
TGA profile of PMA microspheres (A) and ^152^Sm_2_(CO_3_)_3_-PMA microspheres.

**Figure 7 pharmaceutics-15-00877-f007:**
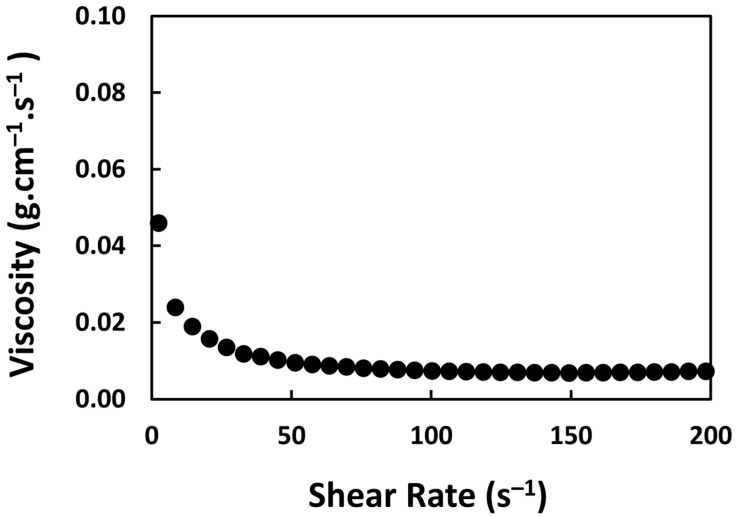
Viscosity of the ^152^Sm_2_(CO_3_)_3_-PMA microsphere suspension at a concentration of 2.5% (*w*/*v*). The viscosity was measured in saline solution of 37 °C at various shear rate.

**Figure 8 pharmaceutics-15-00877-f008:**
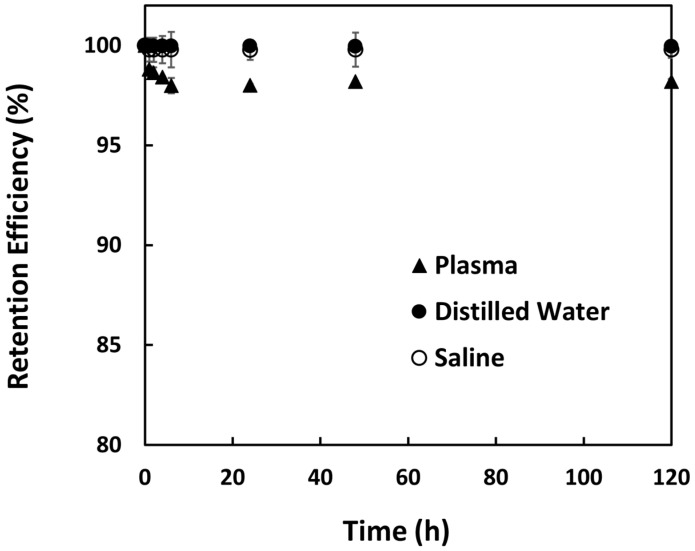
Retention (%) of ^153^Sm in ^152^Sm_2_(CO_3_)_3_-PMA microspheres suspended in distilled water, saline solution and human blood plasma over 120 h.

**Table 1 pharmaceutics-15-00877-t001:** Physicochemical characteristics of the Sm_2_(CO_3_)_3_-PMA microspheres.

Physicochemical Properties	^152^Sm_2_(CO_3_)_3_-PMA Microspheres
Mean size (µm)	29.3 ± 0.18
Density (g·cm^−3^)	1.346 ± 0.0005
Viscosity of 2.5% (*w*/*v*) microsphere suspension at 37 °C (g·cm^−1^·s^−1^)	0.01067 ± 0.0003
Particle concentration (number particles per g)	56 million
Specific activity (GBq·g^−1^)	4.40 ± 0.08
Activity per particles (Bq)	78.6

## Data Availability

No new data were created or analyzed in this study. Data sharing is not applicable to this article.

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
