# Peer review of "Facile Preparation of Samarium Carbonate-Polymethacrylate Microspheres as a Neutron-Activatable Radioembolic Agent for Hepatic Radioembolization"

_pharmaceutics, 2023, doi:10.3390/pharmaceutics15030877_

Round 1

Reviewer 1 Report

General comments:

The paper deals with an important topic, of hepatic cancer management through radioembolization. Embolization via radioisotopes is one of the most promising therapies for these malignancies and more research is required to improve its efficacy. The current paper reports on samarium-based compounds for radioembolization, after neutron activation. The theranostic potential of this isotope is important for its identification in the tumour and for dosimetric evaluation.

The paper is generally well written and scientifically well explained. I would advise the authors to consider an English language revision - some grammar/linguistic errors are listed in the Specific comments, though the list is not exhaustive.

Below are my comments regarding this article:

(1)   The Introduction should include a paragraph justifying the use/efficiency of Sm153 in liver cancers from a radiobiological perspective. The compatibility with neutron cross-section is an advantage, there should be a clinical justification based on physical and radiobiological properties of Sm. Beta emitters in targeted therapies are slowly being replaced by more efficient, alpha emitting isotopes, due to shorter range, higher linear energy transfer and higher relative biological effectiveness. Furthermore, hypoxic tumours benefit much more from alpha-targeted therapies (oxygen enhancement ratio close to 1) that from beta emitters. Please comment on these aspects when justifying the use of Sm.

(2)   Also, are there any in vitro studies on the efficiency of 153Sm on liver cells? If yes, please include such studies in the introduction.

(3)   Add to Discussion a paragraph on further pre-clinical applications of this compound.

Specific comments:

1.     Please be consistent with the terminology: theragnostic or theranostic

2.     Line 71 – add full stop after 30um and delete ‘and’

3.     Line 92 – ‘…using either the pneumatic transfer system…’

4.     Line 97 – ‘The specific activity was then…’ (the specific activity is an allocated terminology and represents the activity of the isotope per unit mass)

5.     Eq 1 – replace ‘specify’ with ‘specific’

6.     Line 121 – ‘The spectra of the samples…’

7.     Line 130 – ‘The density…was measured..’

8.     Line 156 – remove ‘was calculated’ (it appears twice in the sentence)

9.     Figures 4 are not clear. Please improve the quality and magnification of the images.

10.  Line 202 – ‘These peaks remained in the …’

11.  Line 247 – ‘…the positively charged…’

Author Response

We would like to thank Reviewer 1 for providing useful and constructive comments to our manuscript. We believe that these comments have resulted in a significant improved manuscript. The comments from the Reviewer have been addressed in one-to-one basis as stated below. We have corrected the language and style problems as much as possible. Please refer the revised manuscript. Please find enclosed a revised manuscript with this covering letter documenting responses to all the reviewer’s comments. The reply to the comments is in blue font.

Comment 1: The Introduction should include a paragraph justifying the use/efficiency of Sm153 in liver cancers from a radiobiological perspective. The compatibility with neutron cross-section is an advantage, there should be a clinical justification based on physical and radiobiological properties of Sm. Beta emitters in targeted therapies are slowly being replaced by more efficient, alpha emitting isotopes, due to shorter range, higher linear energy transfer and higher relative biological effectiveness. Furthermore, hypoxic tumours benefit much more from alpha-targeted therapies (oxygen enhancement ratio close to 1) that from beta emitters. Please comment on these aspects when justifying the use of Sm.

Reply to Comment 1: We thank the reviewer on the suggestion to improve the introduction. The justification for the use of Sm-153 has been added in Page 1, Lines 71-73, and Page 3, Lines 74-76. The alpha particles therapy is also mentioned in Page 3, Line 82-86.

Comment 2:  Also, are there any in vitro studies on the efficiency of 153Sm on liver cells? If yes, please include such studies in the introduction.

Reply to Comment 2: In-vitro studies on the therapeutic efficiency of Sm-153 have not been carried out, but it has been tested in an in-vivo animal model using liver tumour bearing Sprague-Dawley (SD) rats in a previous study [Tan et al., 2023]. The liver tumour bearing SD rats was divided into two groups, namely, control group (treated with saline solution only) and treatment group (treated with Sm-153 loaded microspheres). The average tumour volume in the control groups increased 24-fold at the end of the study while liver tumour was observed in the treatment group at the end of the study. The result of the study suggested that Sm-153 loaded microspheres demonstrated excellent therapeutic efficiency for liver cancer in the SD rat model. The information has been added in the revised MS (please see page 3, Lines 79-82).

Comment 3: Add to Discussion a paragraph on further pre-clinical applications of this compound.

Reply to Comment 3: The future pre-clinical applications of this compound have been added in page 14, Line 342-346 in revised MS.

Specific Comments:

  1. Please be consistent with the terminology: theragnostic or theranostic
  2. Line 71 – add full stop after 30um and delete ‘and’
  3. Line 92 – ‘…using either the pneumatic transfer system…’
  4. Line 97 – ‘The specific activity was then…’ (the specific activity is an allocated terminology and represents the activity of the isotope per unit mass)
  5. Eq 1 – replace ‘specify’ with ‘specific’
  6. Line 121 – ‘The spectra of the samples…’
  7. Line 130 – ‘The density…was measured..’
  8. Line 156 – remove ‘was calculated’ (it appears twice in the sentence)
  9. Figures 4 are not clear. Please improve the quality and magnification of the images.
  10. Line 202 – ‘These peaks remained in the …’
  11. Line 247 – ‘…the positively charged…’

Reply to Specific Comment: We thank the reviewer for highlighting the grammar/linguistic errors. We have corrected the language and style problems as much as possible. Please refer the revised manuscript.

Reviewer 2 Report

Facile Preparation of Samarium Carbonate-Polymethacrylate 2 Microspheres as a Neutron-Activatable Radioembolic Agent for 3 Hepatic Radioembolization

In the paper under evaluation, the authors reported a rapid method to obtain samarium-PMA microspheres by using neutron activation and a research nuclear reactor. The main advantage of this method is that of providing microspheres labeled with a radionuclide emitting, aside from beta particles, also gamma rays at 103 and 69 keV, suitable for imaging with gamma camera. The authors evaluated the morphology of microspheres before and after the procedure of neutron activation.

The paper is well written, original and interesting. Some considerations:

- In the introduction, the authors write “The treatments deliver microspheres containing chemotherapy or radioactive drugs to the targeted tumour(s) to block the blood supply as well as delivering chemotherapeutics or ionizing radiation to the tumours”. I warmly suggest to add that TARE is preferred in case of HCC with portal vein thrombosis and provided encouraging results in intrahepatic cholangiocarcinoma, also citing the following references DOI: 10.1111/1751-2980.13030, doi: 10.2217/fon-2017-0443

- the authors write “During hepatic radioembolization, the radioembolic microspheres are usually suspended in saline solution at a concentration of 2.5% (w/v) for intraarterial delivery.” Indeed, more recent recommendations suggest to suspend microspheres in 5% glucose solution to reduce stasis. Please amend, also citing doi: 10.1007/s00259-021-05600-z

- the authors state that post radioembolization imaging with 90Y is difficult. In particular, they write: “Bremsstrahlung imaging and Positron Emission Tomography (PET) have been used to verify the treatment outcome, however these techniques suffer from poor spatial resolution and sensitivity, respectively.” I partially agree, 90Y-PET cannot be compared with bremsstrahlung imaging and can provide good quality imaging and reliable quantification, if proper algorithms are used. Please elaborate, also citing: doi: 10.1186/s13550-017-0341-9

Author Response

We would like to thank Reviewer 2 for providing useful and constructive comments to our manuscript. We believe that these comments have resulted in a significant improved manuscript. The reply to the comments is in blue font.

Comment 1: In the introduction, the authors write “The treatments deliver microspheres containing chemotherapy or radioactive drugs to the targeted tumour(s) to block the blood supply as well as delivering chemotherapeutics or ionizing radiation to the tumours”. I warmly suggest to add that TARE is preferred in case of HCC with portal vein thrombosis and provided encouraging results in intrahepatic cholangiocarcinoma, also citing the following references DOI: 10.1111/1751-2980.13030, doi: 10.2217/fon-2017-0443

Reply: We thank the reviewer for the information and suggested references. The information has been added in the revised MS. Please see page 2, Line 45-47.

Comment 2: the authors write “During hepatic radioembolization, the radioembolic microspheres are usually suspended in saline solution at a concentration of 2.5% (w/v) for intraarterial delivery.” Indeed, more recent recommendations suggest to suspend microspheres in 5% glucose solution to reduce stasis. Please amend, also citing doi: 10.1007/s00259-021-05600-z

Reply: We thank the reviewer for highlighting the latest recommendation from EANM. The information has been corrected in the revised MS. Please see the Page 11, Lines 288-290.

Comment 3: the authors state that post radioembolization imaging with 90Y is difficult. In particular, they write: “Bremsstrahlung imaging and Positron Emission Tomography (PET) have been used to verify the treatment outcome, however these techniques suffer from poor spatial resolution and sensitivity, respectively.” I partially agree, 90Y-PET cannot be compared with bremsstrahlung imaging and can provide good quality imaging and reliable quantification, if proper algorithms are used. Please elaborate, also citing: doi: 10.1186/s13550-017-0341-9.

In my opinion, this manuscript seems to be interesting and deserves to be considered for publication in this journal. However, there are some minor revisions that could be implemented to improve the overall merit and quality of the paper, specifically regarding the need to correct some aspect within the theoretical background used to justify the research undertaken, clarify the aims of the work and to discuss deeply some technical issues involved in experiments.

Reply: We thank the reviewer for the insight on the 90Y- based PET imaging. The information has been elaborated in the revised MS. Please see the Page 12, Lines 312-315.
